# PolarStream: Streaming Lidar Object Detection and Segmentation with Polar Pillars

**Qi Chen**[*]
Johns Hopkins University
Baltimore, MD 21218
qchen42@jhu.edu

**Sourabh Vora**
Motional
Santa Monica, CA 90401
sourabh.vora@motional.com

**Oscar Beijbom**
Motional
Santa Monica, CA 90401
oscar.beijbom@motional.com

## Abstract

Recent works recognized lidars as an inherently streaming data source and showed that the end-to-end latency of lidar perception models can be reduced significantly by operating on wedge-shaped point cloud sectors rather then the full point cloud. However, due to use of cartesian coordinate systems these methods represent the sectors as rectangular regions, wasting memory and compute. In this work we propose using a polar coordinate system and make two key improvements on this design. First, we increase the spatial context by using multi-scale padding from neighboring sectors: preceding sector from the current scan and/or the following sector from the past scan. Second, we improve the core polar convolutional architecture by introducing feature undistortion and range stratified convolutions. Experimental results on the nuScenes dataset show significant improvements over other streaming based methods. We also achieve comparable results to existing non-streaming methods but with lower latencies.

## 1 Introduction

The ability to accurately perceive objects in dense urban environments still remains a challenging problem for self-driving cars. While such self-driving cars typically deploy a wide variety of sensors lidars play a key role due to the accurate range information provided. Driven in part by the availability of benchmark datasets [12, 3, 27], the last decade has seen tremendous progress in lidar based 3D object detection [39, 16, 32, 21, 10]. However, these methods all ignore the fact that most lidar sensors scan the scene sequentially as the lidar rotates around the z-axis. They instead wait for the rotational scan to complete (colloquially known as full sweep) before processing data, thereby introducing a large data capture latency (usually 50 to 100 ms).

First, Han et al. [13] and then STROBE [11] recognized this problem and proposed solutions which processed lidar sectors (shown in Fig. 1) as soon as they arrived. They showed that a streaming based architecture can achieve significantly reduced latency over the traditional non-streaming baselines. Both of these methods encode the point clouds as an image in bird's-eye view (BEV) using cuboid-shaped voxels. In doing so, they ignore the natural polar representation formed by the lidar sectors. Using cuboid-shaped voxels restricts them to performing convolutions on the minimal rectangular region enclosing the point cloud sector which wastes both computation and memory. As shown in Fig. 1, a large portion of the enclosed rectangular region remains empty.

---

[*]work done while interning at Motional

35th Conference on Neural Information Processing Systems (NeurIPS 2021).

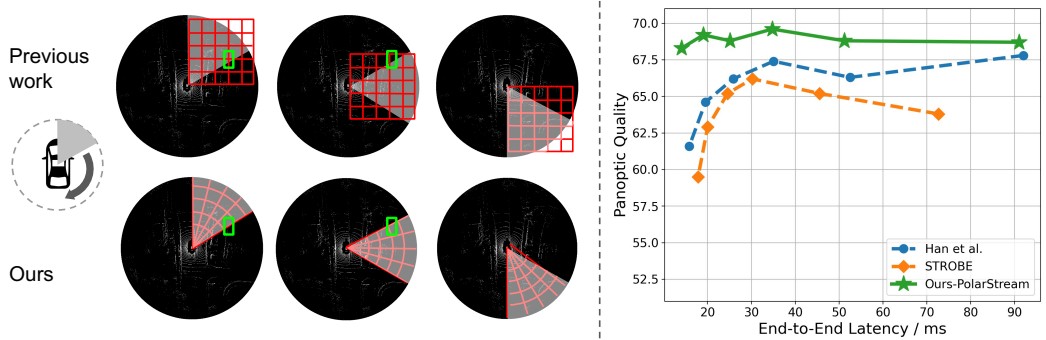

Figure 1: Left: An illustration of streaming lidar point clouds on bird's eye view. Lidar point clouds arrive as wedge-shape sectors (shown in gray masks) as the scanner rotates. Previous methods, Han et al.[13] and STROBE[11], represent the sectors using rectangular regions, wasting half of memory and computation for empty regions. Ours represents the sectors as wedge-shape regions using a polar grid. Right: Comparison of different streaming methods wrt. Panoptic Quality vs End-to-End Lantency as we slice the full sweep into $n = 1, 2, 4, 8, 16, 32$ sectors using the NuScenes[3] val split. The end-to-end latency includes $50/n$ ms for LiDAR scan and the total runtime of the algorithms.

Another challenge associated with streaming perception models is the limited view of the scene observed by each sector. Objects close to the ego-vehicle can often be fragmented across multiple sectors as shown by the car highlighted in green in Fig.1. Han et al.[13] proposes to increase the context available to the model by maintaining a recurrent memory across consecutive sectors. STROBE [11] also aggregates representations from the previous sectors by maintaining full-sweep feature maps across multiple scales. However, both these solutions add extra computation.

In this work, we propose to encode individual point cloud sectors using polar pillars. Polar pillars naturally address the inefficiency of existing streaming approaches by representing the point cloud sectors as more compact wedge-shaped regions as shown in Fig. 1. Further, we propose a simple minimal-latency approach to enhance the context available to the model by simply padding the representation of the neighboring sectors across multiple strides of the backbone. Using polar pillars allows us to pad features from the preceding sector of the current scan and/or the following sector from the previous scan, no matter how many sectors the full sweep is divided into.

The polar BEV representation has recently started gaining attention in the lidar perception literature primarily because it balances the points across grid cells. In fact, polar grid outperforms the cartesian grid on the lidar segmentation task [36, 37]. However, the detection peformance on a polar grid still lags the cartesian grid [1, 5, 23]. This is because of the distortion the objects undergo when this representation is ultimately unfolded to a rectangular representation to enable the use of convolutional layers. The object represented by the green box in Fig. 2 shows an example of this distortion. Further, the distortion increases with range as the pillars progressively become larger. This makes a polar representation not compatible with the translation-invariance property of convolution.

In this work, we propose several techniques to address the distortion problem described above. We first propose a **Feature Undistortion** module which transforms the polar representation into a canonical Cartesian representation (as shown in Fig. 2) for classification branch. Next, we propose using the **Range Stratified Convolution&Normalization** layers on the regression branches of the detection head. These layers apply different convolution kernels and normalization based on range (Fig.2) to cater to the changing pillar sizes in a polar grid. Our proposed model closes the gap on 3D object detection models using cartesian representations without adding any significant latency.

Finally, we train multitasking streaming models that do simultaneous 3D object detection, lidar segmentation and panoptic segmentation, for the first time in literature. Results on the nuScenes dataset show that our proposed model **PolarStream** outperforms all streaming methods in both panoptic quality and speed. PolarStream also stays competitive with the top-performing lidar perception methods on the nuScenes leaderboard while being at least twice as fast as the rest. We do several ablation studies and extensive analysis to show the effectiveness of PolarStream.

In summary, our contributions are:

- An efficient streaming based lidar perception models using a polar grid.
- Multi-scale context padding: an efficient approach to enhance the context of streaming lidar perception models
- Several improvements to the core problem of applying convolutions on a polar grid: Feature Undistortion, Range Stratified Convolution&Normalization all add minimal latency to our model.

## 2 Related Works

### 2.1 Non-streaming lidar perception

Most lidar perception architectures take inspiration from the image perception literature [24, 18, 17]. Some single-stage methods typically convert the point cloud into a bird's-eye view image [39, 16, 32] or a range view image [21, 10] and perform detection in those views. The most common paradigm is to convert the lidar point cloud into a BEV image as it offers several advantages like a lack of scale ambiguity, a near lack of occlusion, the ease of fusing HD maps [31] and performing simultaneous detection and trajectory predictions [4, 19]. To convert the point clouds into a BEV representation, most existing models choose to group the points into voxels. The most commonly used voxels are cuboid-shaped based on Cartesian coordinates. VoxNet [20], MV3D [7], Pixor [33], Complex-YOLO [26] represent the cuboid-shaped voxels as occupancy grids. To avoid quantization effects of occupancy grids and extract richer voxel features, VoxelNet [39] samples a fixed number of points within each voxel and applies a simple PointNet [22] to them. For efficiency, PointPillars [16] discretizes the 3D space into pillars so there is only one voxel along the height dimension.

Some recent methods that operate on BEV start to explore polar voxels for point clouds. For 3D object detection, Alsfasser et al [1] voxelizes points under the Cylindrical Coordinate System, MVF [38] adopts both cuboid-shaped voxels and spherical voxels, and CVCNet combines cylindrical and spherical coordinate system into one Hybrid-Cylindrical-Spherical (HCS) coordinate system to detect object from both bird's eye view and range view. On the other hand, the success of PolarNet [36] and Cylinder3D [37] shows the advantage of Cylindrical grids over Cartesian voxels in LiDAR semantic segmentation. Panoptic-PolarNet [40] further extends PolarNet to the task of panoptic segmentation.

### 2.2 Streaming lidar perception

Streaming lidar perception is relatively new in literature and offers a compelling argument in reducing the end-to-end latency. Han et al [13] proposed a couple of enhancements to convert a 3D object detector to operate on streaming data: a) using an LSTM to accumulate features from preceding sectors and b) applying stateful NMS to suppress objects across multiple sectors. STROBE [11] accumulates features not only from the preceding sectors of the same scan but also from the previous scan by maintaining multi-scale memory feature maps. Features extracted from the current sector is concatenated and fused with the corresponding cropped region in the memory feature maps.

## 3 PolarStream

In this section, we introduce PolarStream, a streaming model based on polar pillars. We introduce how we prepare lidar streaming data in Sec.3.1, polar pillars as a representation for point clouds sectors in Sec.3.2, the simultaneous detection and segmentation model including techniques to improve detection on a polar grid in Sec.3.3, and multi-scale context padding to enlarge context in Sec.3.4.

### 3.1 Streaming LiDAR Inputs

Since there is no streaming lidar dataset available, we simulate a streaming system from the NuScenes dataset [3] by slicing the point clouds into n sectors according to their azimuth. As shown in Fig.1, each sector is like a slice of a full pizza. We try $n = 1, 2, 4, 8, 16, 32$ sectors in our experiments, where $n = 1$ means full sweep. The dataset contains $1,000$ scenes, comprising 700 scenes for training, 150 scenes for validation and 150 scenes for test. Each scene is of $20s$ duration, captured by 32-beam lidar. $40,000$ frames are annotated in total, including 10 object categories such as cars, motorcycles and pedestrians and six stuff classes such as vegetation and drivable region. We consider 10 object classes for detection, 16 classes in total for semantic and panoptic segmentation.

### 3.2 Polar Pillars

The point clouds sector consists of $N$ points, each represented by a vector of point feature $f_p = (r_p, \theta_p, z_p, x_p, y_p, i_p, t_p)$, where $(x_p, y_p, z_p)$ is its Cartesian coordinates. $(r_p, \theta_p)$ is the polar coordinates. $i_p$ is the reflection intensity and $t_p$ is the timestamp when the lidar point is captured. Points are accumulated from 10 successive frames in total to obtain denser point clouds. The points from previous frames are motion-compensated and transformed to current frame. We group the

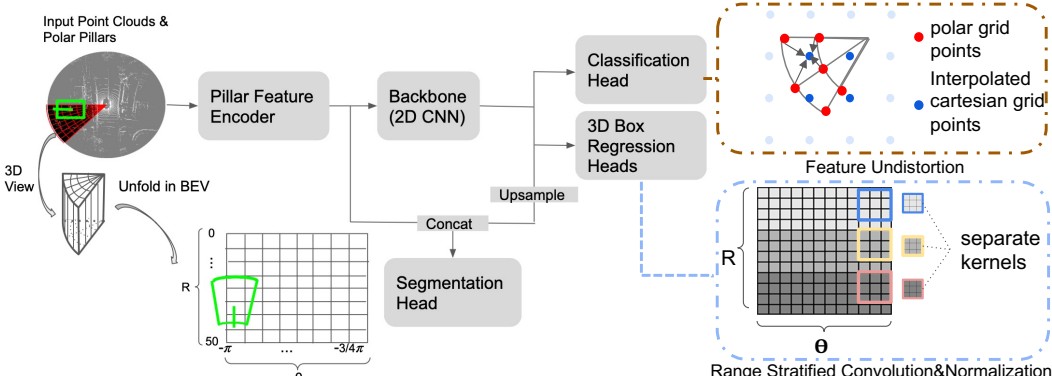

Figure 2: Simultaneous LiDAR object detection and segmentation network with polar pillars. We adopt the same backbone as in PointPillars[16], and add a semantic segmentation head in parallel with the detection heads. The input wedge-shape pillars are unfolded into a rectangular feature map for convolution. The object (green box) is distorted because one end near the sensor looks bigger and the other end far from the sensor looks smaller. Feature Undistortion is applied to classification head to mimic bilinear sampling and interpolate cartesian pillar features from polar pillar features. Range Stratified Convolution& Normalization is applied to center offset regression head.

points according to the cylindrical pillar resolution $(\delta r, \delta\theta, \delta z)$ where $\delta z = z_{max} - z_{min}$ so there is only one pillar along the height dimension. Following MVF [38], we adopt dynamic voxelization to sample all points within each pillar, instead of randomly sampling a fixed number of points per pillar.

### 3.3 Simultaneous Detection and Segmentation

We design PolarStream: a simultaneous object detection and segmentation network by extending PointPillars [16], one of the most widely used 3D object detectors balancing accuracy and speed. As shown in Fig.2, PolarStream consists of a Pillar Feature Encoder, followed by a 2D CNN backbone and a U-Net[25] like structure. On top are the detection and segmentation heads.

**Detection Heads**  We adopt CenterPoint [35] heads with modifications to make it compatible with polar pillars. To assign targets to the 10-class heatmap to indicate the objects, the gaussian radius of the object center is computed using the span of range and azimuth of the object bounding box, instead of using length and width of the box. Following CenterPoint, we also regress the center offset as $d_x, d_y$, the bounding box size $l, w, h$ as $\log l, \log w, \log h$, and predict the bounding box height $z$. We regress the relative bounding box orientation $\phi$ as $\cos\phi, \sin\phi$ and relative velocity as $v_x, v_y$ similar to [23]. Unlike most methods, which use multi-group detection heads that partition object classes to several groups according to their size, we use single-group detection heads to balance accuracy and speed. A comparison against multi-group detection heads is shown in Supplementary. For streaming data with $n > 1$, we apply stateful-NMS proposed in Han et al.[13].

**Segmentation Head**  To extend PointPillars for segmentation, we add a semantic segmentation head in parallel with the detection heads. The segmentation head is made of a single 1x1 convolution layer. The input for the segmentation head is concatenation of the outputs from pillar feature encoder and bilinearly upsampled features from the 2D backbone.

**Panoptic Fusion**  Similar to Panoptic-PolarNet [40], for each point belonging to things, we predict the instance id as the box id whose category is the same and center is the nearest. For streaming data with $n > 1$, the panoptic segmentation task is not well defined. For example, the points in the $(i)_{th}$ sector may belong to the box in the $(i + 1)_{th}$ sector if the majority of the box is in the $(i + 1)_{th}$ sector. However, when we are doing panoptic fusion for $i_{th}$ sector, we do not have information from the $(i + 1)_{th}$ sector. Therefore we choose global panoptic fusion for streaming point clouds, i.e., we assign instance ids according to the boxes from all sectors of the same sweep.

**Multi-Task Learning**  We adopt Focal Loss [17] for classification and L1 loss for bounding box regression, orientation and velocity estimation. For segmentation, we use the weighted cross-entropy loss and lovasz-softmax loss [2]. The total loss is the weighted sum of losses for each component.

**Feature Undistortion**  As mentioned in Sec.1, objects have distorted appearances with polar pillars, we propose Feature Undistortion to undistort the features. As shown on the top right of Fig.2, the idea of undistortion is to interpolate features at cartesian pillar locations from the original polar pillar

locations so that the translation-invariant property of convolution applies. We find the connection of bilinear sampling to convolution and mimic bilinear sampling using convolution. For bilinear sampling, the interpolated features at point $p$ can be sampled from its neighboring points $\mathcal{N}_p$:

$$f_p = \sum_{p_k \in \mathcal{N}_p} w_k f_{p_k} \tag{1}$$

where $w_k$ is a function of distance$(p, p_k)$.

We find Equation 1 has the similar form to convolution, except that for convolution $w_k$ is fixed because same kernel is slided through every location of the feature map. To make $w_k$ distance-dependent, we tweak Equation 1 by adding a new parameter $w'_k$ so Equation 1 can be rewriten as:

$$f_p = \sum_{p_k \in \mathcal{N}_p} w_k w'_k f_{p_k} \tag{2}$$

where $w'_k$ is conditioned on distance$(p, p_k)$. We model $w'_k$ as the output of a neural network. We build a standalone fully convolutional network $g$ that takes position encodings at $p_k$ and its neighboring points $\mathcal{N}_{p_k}$, i.e. $\{pe_i = (r_i, \cos\theta_i, \sin\theta_i, x_i, y_i) | i \in \mathcal{N}_{p_k} \cup p_k\}$ as input, and output $w'_k$. Simply put:

$$w'_k = g(\{pe_i\}) \tag{3}$$

To make it more general, we also add a bias term $b'_k$, and another standalone network $q$ so that $b'_k = q(\{pe_i\})$ and

$$f_p = \sum_{p_k \in \mathcal{N}} w_k(w'_k f_{p_k} + b'_k) \tag{4}$$

$g$ and $q$ is trained together with our main network, and during inference $w'_k$ and $b'_k$ are fixed for each location $p_k$ so it does not need extra runtime for $g$ and $q$. We apply feature undistortion in center heatmap prediction.

**Range Stratified Convolution&Normalization** Another challenge with polar pillars is that the center offset is dependent on range and azimuth so it has different statistics at different regions: suppose the heatmap center is at $(r_c, \theta_c)$, and the target is at $(r_t, \theta_t)$. The center offset is

$$d_x = r_t \cos\theta_t - r_c \cos\theta_c \quad d_y = r_t \sin\theta_t - r_c \sin\theta_c \tag{5}$$

For simplicity, assume $r_t = r_c$, i.e. the center offset moves along a circle. Suppose $\theta_t > \theta_c$ then

$$\theta_t = \theta_c + \theta_s < \theta_c + \delta\theta \tag{6}$$

where $\theta_s$ is a small angle and $\delta\theta$ is the polar pillar angle size. Then

$$d_x = r_c(\cos(\theta_c + \theta_s) - \cos\theta_c) \approx -r_c\theta_s \sin\theta_c \tag{7}$$

Similarly, we can derive that $d_y$ is also dependent on range and azimuth and observe that for Cartesian pillars center offset ranges from -1 to 1 and mean is 0.49 and std is 0.28, while polar pillars center offset ranges from -2 to 2, mean is 0 and std is 0.64. The polar std is much larger than that for Cartesian pillars. Hence it's more difficult to regress center offset based on polar pillars. Based on these observations, we propose Range Stratified Convolution& Normalization instead of regular convolution and batch normalization[15]. As shown on bottom right of Fig.2, Range Stratified Convolution applies individual kernels at different ranges and Range Stratified Normalization only normalizes over individual regions within certain range instead of entire spatial dimension. We apply Range Stratified Convolution&Normalization to center offset regression. We also apply Range Stratified Normalization to the shared convolution for detection heads.

### 3.4 Multi-Scale Context Padding

**Trailing-Edge Context Padding** As shown in Fig.3, the sector is unfolded to a rectangle feature map on $r$-$\theta$ plane as input for convolution. The lidar sectors arrive one after another by increasing the angle the sensor scans so the unfolded feature map of a sector is spatially connected to its preceding sector along $\theta$ dimension. This unique property of using polar pillars inspires us to, instead of zero-padding along $\theta$ dimension, pad the features from preceding sector where it is spatially connected to current sector. The receptive field of a neuron increases as the neural network goes from bottom layer to top and the network encodes multi-scale representation of the input at different stages. This motivates us to pad context from preceding sector before every convolution of the 2D CNN backbone, as illustrated in trailing-edge padding of Fig.3. Although we only pad a few columns to the feature map, the neural network is replenished with sufficient context from multiple ranges and multiple scales at different stages of the network. We keep zero-padding for $r$ dimension and the other end of the $\theta$ dimension, as the other end of $\theta$ dimension points to the future sector.

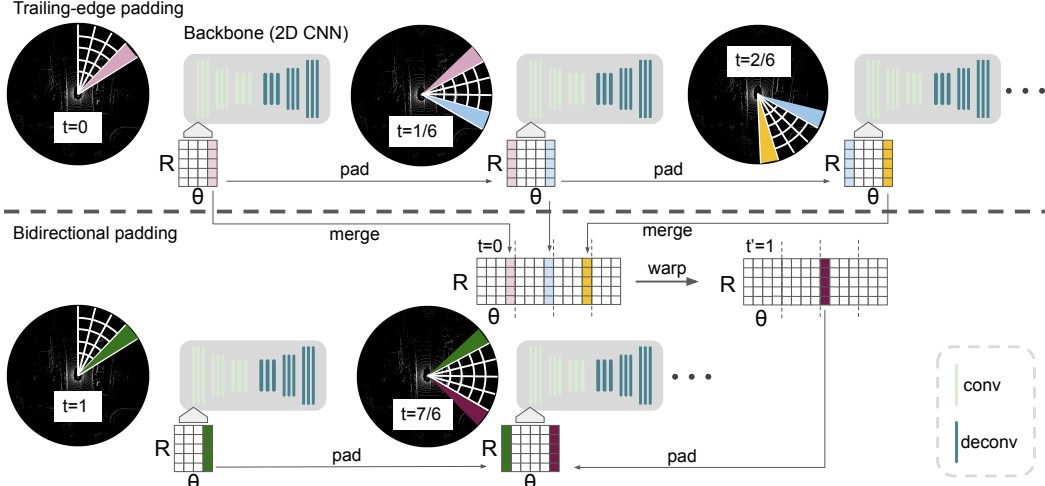

Figure 3: Multi-Scale Context Padding. We present both trailing-edge padding and bidirectional padding. Trailing-edge padding pads current sector with features from preceding sector. Bidirectional padding additionally pads current sector with features from 'following' sector of past time frame. Full-sweep feature maps are merged for past time frame and warped to the coordinate system of current time by ego-motion compensation. Context Padding is applied to every convolution in the backbone.

**Bidirectional Context Padding**  With trailing-edge padding the current sector is padded with context from preceding sector. To provide further context we pad the leading-edge with warped features from the following sector of the *previous* sweep. To do this we aggregate the full-sweep multi-scale feature maps from the previous sweep and warp the feature maps to the coordinate system of current sweep using ego-motion compensation. We then pad the leading edge of the current sector with the corresponding warped features spatially connected to the current sector.

## 4    Implementation

### 4.1    Network Details

For polar pillars with $n$ sectors per sweep, $r, \theta, z$ range is $[0.3, 50.3]$m, $[-3.1488, -3.1488 + 6.2976/n]$ rad and $[-5, 3]$m, the pillar size is $(0.098, 0.0123, 8)$. For Cartesian pillars, the pillar size is $(0.2, 0.2, 8)$. When $n = 1$, $x, y, z$ range is $[-51.2, 51.2]$m, $[-51.2, 51.2]$ m and $[-5, 3]$m, leaving same input size of $512 \times 512 \times 1$ for both Cartesian pillars and polar pillars when $n = 1$. We find the minimal rectangular region to enclose the sectors when $n > 1$ for Cartesian Pillars. We set segmentation loss weight to 2 and classification loss to 1 for both polar pillars and Cartesian pillars. For Cartesian pillars the bounding box regression weight is 0.25. For polar pillars, since regression is harder, we set the loss weight to 0.5. We make sure they are the best configuration for each setting. For $g$ and $q$ in Feature Undistortion, they share the same architecture: a 3x3 conv followed by 1x1 conv with tanh as activation. We show the network architecture in Supplementary. All runtimes are measured on a single V100 GPU using Pytorch.

### 4.2    Augmentation

We adopt class-balanced sampling as proposed in CBGS [41]. Before slicing the point clouds into sectors, we conduct random flipping along $x, y$ axes, scaling with a scale factor sampled from [0.95, 1.05], rotation around $z$ axis between [-0.3925, 0.3925] rad and translation in range $[0.2, 0.2, 0.2]$ m in $x, y, z$ axis. Unlike most methods, we do not use database sampling[30] for fast training.

## 5    Experiments and Results

### 5.1    Evaluation

We gather the predictions from individual sectors and evaluate PolarStream similar to full-sweep methods. We evaluate 3D detection and lidar semantic segmentation on the NuScenes benchmark [3]. The detection mean average precision (mAP) is based on the distance threshold (i.e. $0.5m, 1.0m, 2.0m$ and $4.0m$). Additionally, we use nuScenes detection score (NDS) [3], a weighted sum of mAP and

Table 1: Comparison of streaming methods on nuScenes Val split. CP: context padding; CP x1: trailing-edge padding; CP x2: bidirectional padding.

| Method | Panoptic Quality (PQ) | | | | | | Segmentation mIoU | | | | | |
|---|---|---|---|---|---|---|---|---|---|---|---|---|
| | 1 | 2 | 4 | 8 | 16 | 32 | 1 | 2 | 4 | 8 | 16 | 32 |
| Cartesian Pillars | 67.8 | 66.8 | 67.6 | 65.5 | 64.1 | 59.8 | 72.1 | 71.5 | 70.9 | 70.2 | 68.6 | 65.7 |
| Han et al.[13] | - | 66.3 | 67.4 | 66.2 | 64.6 | 61.6 | - | 70.6 | 71.5 | 70.1 | 69.2 | 66.5 |
| STROBE[11] | 63.8 | 65.2 | 66.2 | 65.2 | 62.9 | 59.5 | 69.2 | 69.7 | 69.8 | 69.6 | 67.6 | 65.6 |
| Polar Pillars | **68.7** | 68.6 | 67.7 | 66.2 | 63.9 | 60.3 | **73.4** | 73.4 | 72.5 | 70.8 | 69.5 | 67.1 |
| Ours-PolarStream | - | **68.8** | **69.6** | **68.8** | **69.2** | **68.3** | - | **73.5** | **74.2** | **73.4** | **73.8** | **73.1** |

| Method | Detection mAP | | | | | | Detection NDS | | | | | |
|---|---|---|---|---|---|---|---|---|---|---|---|---|
| | 1 | 2 | 4 | 8 | 16 | 32 | 1 | 2 | 4 | 8 | 16 | 32 |
| Cartesian Pillars | **52.3** | 51.3 | **54.9** | 49.7 | 52.4 | 47.6 | **60.7** | 60.3 | **62** | 57.9 | 59.5 | 57.1 |
| Han et al.[13] | - | 50.9 | 52.9 | **53.8** | 52.7 | 50.6 | - | 59.6 | 60.3 | 60.8 | 60.3 | 58 |
| STROBE[11] | 46.9 | 48.7 | 49.4 | 47.9 | 45.4 | 42 | 53.8 | 54.6 | 51.5 | 48.9 | 47.1 | 44.8 |
| Polar Pillars | 51.2 | 52.1 | 51.9 | 52.5 | 51.9 | 46.7 | 60 | **61.4** | 61.4 | 60.9 | 60.3 | 55.8 |
| Ours-PolarStream | - | 51.5 | 53.2 | 52.7 | **53.9** | **52.4** | - | 60.3 | 61.2 | **60.9** | **61.4** | **60** |

precision on box location, scale, orientation, velocity and attributes. For semantic segmentation, we follow the standard mean intersection-over-union (mIoU) metric. Since nuScenes does not provide instance labels for panoptic segmentation, we follow Panoptic-PolarNet [40] to generate labels and evaluate panoptic segmentation on validation split using the Panoptic Quality (PQ) metric.

## 5.2 Comparison with other Streaming Methods

**Baselines**  Han et al.[13] and STROBE[11] did not release their code and in addition performed evaluation on two different datasets. To enable benchmarking we re-implemented their methods using the same backbone and input resolution as we use and evaluated on the nuScenes dataset. Specifically we re-implemented stateful-NMS and stateful-RNN of Han el al. and multi-scale memory module in STROBE. We did not implement the HD map branch in STROBE in order to ensure a fair comparison. We also apply stateful-NMS and global panoptic fusion to all the methods in comparison as they are just post-processing techniques. We extend both methods to the task of simultaneous object detection, semantic segmentation and panoptic segmentation. We also provide baselines that simply apply Cartesian pillars or polar pillars to individual point clouds sectors. We compare panoptic quality, segmentation mIoU, detection mAP, NDS with the baselines using $n = 1, 2, 4, 8, 16, 32$ sectors (Tab.1). We also show the comparison of our method to Han et al. and STROBE wrt. PQ vs. end-to-end latency (Fig. 1).

**Results**  Tab. 1 shows that PolarStream outperforms all previous streaming methods, including the Cartesian pillars baseline, in both PQ and Segmentation mIoU. When there are more than four sectors in a scene, previous methods as well as the Cartsian/polar pillar baselines show a trend of decreasing PQ and mIoU as the number of sectors increases. However, our PolarStream with bidirectional context padding does not show such a trend: the performance remains almost the same or even better than the full-sweep method. When $n = 1$, PolarStream got +0.9 and +1.3 improvement in PQ and segmentation mIoU compared to the Cartesian pillars baseline. When sectors become smaller and spatial context becomes limited, the improvement is more significant. When $n = 32$, our PolarStream with Bidirectional Context Padding outperforms all previous streaming methods by a large margin, with +6.7 and +6.6 improvements in PQ and segmentation mIoU. This shows that our bidiretional context padding makes better use of spatial context compared to previous methods. Our detection NDS is always the highest or at least on par with the highest among all streaming models. Interestingly, Cartesian pillars/Han et al.'s method show higher mAP than ours for 1, 4 and 8 sectors, when the sectors have plenty of spatial view and our context padding does not have many benefits. Our PolarStream outperform all previous streaming methods in detection mAP for 16 and 32 sectors, when spatial view is limited and Bidirectional Context Padding shows more advantages. In addition, the orientation and velocity error of Han et al.'s is on average 14.6% and 12.9% higher than ours, which will cause problems for the downstream tracking and prediction tasks. As shown in Fig.1, our

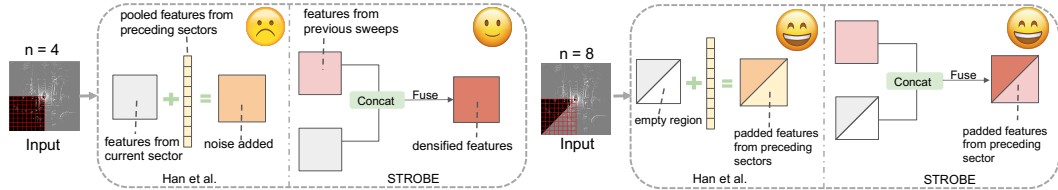

Figure 4: An illustration of how previous streaming methods, Han et al.[13] and STROBE[11] enlarge spatial context of current sector.

methods offers better operating points considering both accuracy and end-to-end latency. Detailed metrics including velocity error and per-class metrics are shown in Supplementary.

## 5.3 Discussions on Streaming

**Full Sweep vs. Streaming**  Contrary to the findings of Han et al[13], we saw improved detection performance using 2, 4, 8 and 16 sectors as compared to models trained on full-sweeps. We hypothesize this improvement to less variation in point coordinates within a sector since all sectors are first transformed to a canonical coordinate frame before processing. This suggests that simulating streaming lidars can also serve as an augmentation technique for full-sweep detection.

**Diagnosis of Previous Streaming Methods**  We first analyze STROBE's low performance in Tab. 1. While all other methods aggregate points from past 10 sweeps after ego-motion compensation (*Point Warp*), STROBE processes the points one sweep at a time, and aggregates information from the past sweeps by first transforming the features based on ego-motion (*Feature Warp*) and then fusing them with the current frame. As shown in the full-sweep case in Tab.1, all the metrics for STROBE are significantly lower than other methods. We thus find Feature Warp inferior to Point Warp for detection and especially for velocity estimation. The average velocity error (AVE)[3] of STROBE is 0.607 m/s, significantly higher compared to Cartesian Pillars with Point Warp (0.358 m/s). We speculate that this high velocity error is because the feature maps in STROBE don't encode time information on account of processing one sweep at a time as compared to the other methods which encode the time lag for each accumulated point from the past 10 sweeps. For 1, 2, 4 sectors, STROBE enjoys the lowest latency because it processes fewer points compared to other methods, also shown in Fig. 1. The pillar feature encoder runs faster. But this advantage disappears for more than 8 sectors because there are also only a smaller number of points processed by all other methods.

Compared to baseline Cartesian Pillars, the method of Han et al. only start to work when more than 8 sectors per sweep. For 2 and 4 sectors, it even hurts the accuracy especially in object detection. This can be explained in Fig. 4. For 2 and 4 sectors, the feature map is fully occupied. Adding the pooled features from preceding sectors is like adding noise to current sector, resulting in worse accuracy. Starting from 8 sectors, there is an empty region in current sector so adding the pooled features from preceding sectors is like padding the empty region, and therefore enlarging the context.

**How Streaming Models Enlarge Context**  We further hypothesize not only our method and Han et al work by padding context from preceding sectors, but also STROBE works by padding. As shown in Fig. 4, for 2 and 4 sectors when the feature map is fully occupied, fusing features from previous sweep is like densifying the features. Starting from 8 sectors, when there is an empty region, fusing features is like padding the empty region. We argue that all existing streaming methods work by padding, but in different format. STROBE and Han et al. are restricted by the shape of the sectors and require empty region as placeholder, and the padded features are added or fused to the placeholder. Our method pads along the edges of feature maps and is not constrained by the shape of the sector.

**Further Thoughts about Context**  We argue that context has two aspects. First is its feature values, for the texture information it carries. Second is its spatial relation to the object of interest. Since convolution is translation-invariant, convolutional neural networks alone do not encode spatial relation. The spatial relation is maintained in the spatial arrangement of neurons on the feature map. The stateful-RNN in Han et al. must work together with the empty region as placeholder to maintain the spatial arrangement, while stateful-RNN alone does not encode spatial relation. On the other

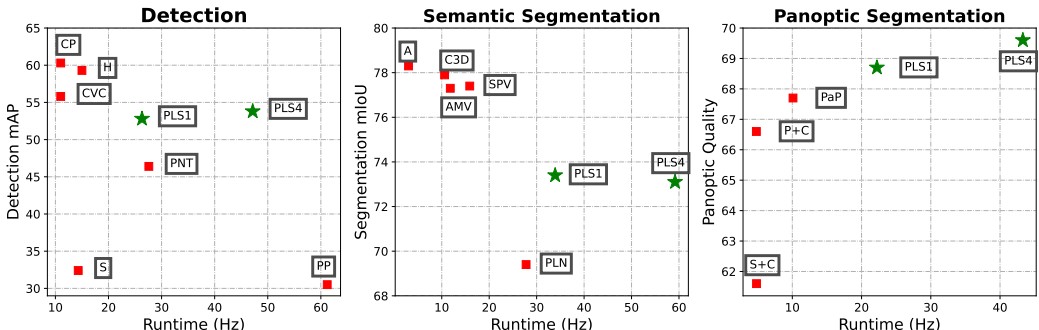

Figure 5: Comparison of our methods, full-sweep PolarStream (PLS1) and PolarStream with Bidirectional Context Padding using four sectors (PLS4), and other methods on the nuScenes dataset. We compare detection and semantic segmentation on the nuScenes benchmark, and panoptic segmentation on the nuScenes val split. We only compare with methods that report both accuracy and runtime. The methods in comparison are: CenterPoint[35](CP), HotSpotNet[6](H), CVCNet[5](CVC), PointPainting[29](PNT), PointPillars[16](PP),SAPNET[34](S), AF2S3Net[8](A), Cylinder3D[37](C3D), PolarNet[36](PLN),SPVNAS[28](SPV), SalsaNext[9]+CBGS[41](S+C), PolarNet+CBGS(P+C) and PaP[40](PaP). We color our methods in green and other methods in red.

Table 2: Comparison with state-of-the-art methods on nuScenes Val split.

| Method | 3D Detection | | | Semantic Segmentation | | |
|---|---|---|---|---|---|---|
| | mAP | runtime(Hz) | #parameters(MB) | mAP | runtime(Hz) | #parameters(MB) |
| CenterPoint[35] | 56.4 | 11 | 149 | - | - | - |
| Cylinder3D[37] | - | - | - | 76.1 | 11 | 215 |
| Ours-PLS1-heavy | 56.2 | 11 | 149 | 76.8 | 15 | 149 |

hand, although our padding along the feature map edges seems simple, it is an effective solution to both add feature values and maintain spatial relation of context.

## 5.4 Comparison with other Full-Sweep Models

As the full-sweep 3D object detection and LiDAR semantic segmentation have longer histories compared to streaming models, there are more full-sweep methods in the literature. We also compare with these methods. We present the results of our full-sweep PolarStream model with $n = 1$ (PLS1) and best performing PolarStream model with $n = 4$ (PLS4), with the same backbone as in PointPillars[16]. As shown in Fig. 5, our method maintains a good balance of runtime and accuracy compared to other methods on both the nuScenes detection and semantic segmentation benchmark. We achieve even faster runtime with PLS4 while preserving almost same accuracy as PLS1. The panoptic segmentation results on the nuScenes val split show that our methods outperform all existing methods in PQ with at least $55\%$ less runtime.

We also adopt a heavier 3D ResNet[14] backbone as in CBGS[41] and compare with the state of the art methods for 3D object detection and semantic segmentation in Tab. 2. Our PLS1-heavy is able to match/beat the state-of-the-art models for detection (CenterPoint[35]) and segmentation (Cylinder3D[37]) on the nuScenes validation set. In this work, we focus on onboard applications so we only choose the same backbone as in PointPillars[16] for streaming.

## 5.5 Ablation Studies

**The Effect of Multi-Scale Context Padding** As shown in Tab.3, the advantage of Multi-Scale Context Padding starts to show up for 8, 16 and 32 sectors, especially in segmentation. For 32 sectors, when the spatial view is the most restricted, we observe the largest gain in detection mAP and segmentation mIoU. Multi-Scale Context Padding does not improve or hurt the baseline polar pillars model for 2 and 4 sectors, because the network already sees enough spatial view. As we increase the amount of context, from trailing-edge padding to bidirectional padding, we observe

Table 3: Ablation Study of Multi-Scale Context Padding on nuScenes Val split. CP: context padding; CP x1: trailing-edge padding; CP x2: bidirectional padding.

| Method | Panoptic Quality (PQ) | | | | | | Segmentation mIoU | | | | | |
|---|---|---|---|---|---|---|---|---|---|---|---|---|
| | 1 | 2 | 4 | 8 | 16 | 32 | 1 | 2 | 4 | 8 | 16 | 32 |
| Polar Pillars | 68.7 | 68.6 | 67.7 | 66.2 | 63.9 | 60.3 | 73.4 | 73.4 | 72.5 | 70.8 | 69.5 | 67.1 |
| Ours-w/ CP x1 | - | 68.6 | 69 | 68.1 | 66.4 | 63.4 | - | 73.3 | 73.7 | 72.6 | 71.6 | 70 |
| Ours-w/ CP x2 | - | **68.8** | **69.6** | **68.8** | **69.2** | **68.3** | - | **73.5** | **74.2** | **73.4** | **73.8** | **73.1** |

| Method | Detection mAP | | | | | | Detection NDS | | | | | |
|---|---|---|---|---|---|---|---|---|---|---|---|---|
| | 1 | 2 | 4 | 8 | 16 | 32 | 1 | 2 | 4 | 8 | 16 | 32 |
| Polar Pillars | 51.2 | 52.1 | 51.9 | 52.5 | 51.9 | 46.7 | 60 | **61.4** | **61.4** | 60.9 | 60.3 | 55.8 |
| Ours-w/ CP x1 | - | **52.2** | **53.6** | 52.4 | 52.3 | 49.3 | - | 60.3 | 61.2 | 60.8 | 60.4 | 58.8 |
| Ours-w/ CPx2 | - | 51.5 | 53.2 | **52.7** | **53.9** | **52.4** | - | 60.3 | 61.2 | **60.9** | **61.4** | **60** |

more improvements. Surprisingly, we observe that with 2, 4, 8 and 16 sectors, our PolarStream with bidirectional padding even outperforms our full-sweep baseline of polar pillars in all metrics including PQ, detection mAP, NDS and segmentation IoU. It suggests that streaming models can be both faster and more accurate.

**The effect of Feature Undistortion and Range Stratified Convolution& Normalization**     We do the ablation studies with $n = 1$, i.e., the full-sweep case. To show how we close the gap of detection accuracy between polar pillars and Cartesian Pillars, we also list the results of Cartesian pillars with the same architecture and input size. We find that polar pillars outperforms Cartesian pillars in semantic segmentation mIoU (73.2 vs 72.1), which is also found in prior arts[36], because points in the same polar pillar have less disagreement in the semantic label compared to those in a Cartesian pillar. However, polar pillars is less accurate in object detection due to the challenges we discussed. In Tab. 4 we show either Range Stratified Convolution& Normalization or Feature Undistortion helps to improve detection accuracy based on polar pillars (by 0.9 and 0.4 mAP respectively). With both techniques combined, we improve detection mAP from $48.2$ to $50.3$, narrowing the gap compared to Cartesian pillars ($50.6$). We also apply both techniques to Cartesian pillars and they do not improve Cartesian pillars, showing they only address the specific challenges of polar pillars, instead of improving the performance by adding more parameters to the network. Our techniques do not add noticeable runtime (0.5ms). In addition, we find detection mAP can be improved when simultaneouly trained with semantic segmentation. The improvement is more significant for Cartesian pillars, but Cartesian pillars suffer from slight drop in segmentation mIoU.

Table 4: Ablation Studies on the validation split of nuScenes.

| Method | seg | det | Stratified Norm&Conv | Feature Undistortion | Runtime (ms) | Det mAP | Seg mIoU |
|---|---|---|---|---|---|---|---|
| Polar Pillars | ✓ | | | | 29.5 | - | 73.2 |
| | | ✓ | | | 38.2 | 48.2 | - |
| | | ✓ | ✓ | | 38.4 | 49.1 | - |
| | | ✓ | | ✓ | 38.4 | 48.6 | - |
| | | ✓ | ✓ | ✓ | 38.7 | 50.3 | - |
| | ✓ | ✓ | ✓ | ✓ | 44.9 | 51.2 | 73.4 |
| Cartesian Pillars | ✓ | | | | 29.8 | - | 72.5 |
| | | ✓ | | | 38.1 | 50.6 | - |
| | | ✓ | ✓ | ✓ | 38.5 | 50.6 | - |
| | ✓ | ✓ | | | 44.5 | 52.3 | 72.1 |

# 6   Conclusion

In this work we propose a streaming model for simultaneous 3D object Detection, Lidar Segmentation and Panoptic Segmentation. Polar pillars is introduced as a more compact representation for lidar sectors compared to previous methods. Multi-scale context padding including trailing-edge padding and bidirectional padding is proposed to enhance spatial context of the streaming model with minimal latency. Additionally we make several improvements, Feature Undistortion and Range Stratified Convolution& Normalization, to address the problem of applying convolutions on a polar grid. Our model showed significant improvements over previous streaming methods with lower latency.

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
