# Supplementary Materials for PolarStream

In this document we present additional ablation studies in Sec.1, 2 and 3. We describe implementation details in Sec.4. We show detailed results including per-class metrics in Sec.5 and 6. The memory usage comparison between Cartesian pillars and polar pillars for streaming is presented in Sec. 7.

## 1   The Effect of Different Panoptic Fusion Methods for Streaming

In the main paper we present global panoptic fusion, i.e., considering detected object bounding boxes from all sectors at the same sweep when assigning instance ids. Here we also present stateful fusion: considering detected boxes from previous and current sectors. Additionally we show the results when each fusion method is combined with stateful NMS [1], i.e. instead of doing NMS within current sector, doing NMS considering boxes in current sector and previous sectors. We show the results of our PolarStream with bidirectional padding. As shown in Tab.1, with stateful fusion the panoptic quality for streaming drops significantly especially for 16 and 32 sectors. Stateful fusion is not reasonable for streaming because points from multiple sectors may belong to the same object, while the model can not see detected boxes from the following sectors with stateful fusion. We also find adding stateful-NMS improves panoptic quality because stateful-NMS suppress false positives of current sector by detected boxes from previous sectors, while pantoptic quality is very sensitive to false positives. When combining stateful NMS and global fusion the pantopic quality of streaming models is almost the same or even better than the full-sweep method (68.7).

Table 1: Comparison of panoptic fusion methods on nuScenes Val split.

| Method | Panoptic Quality (PQ) | | | | |
|---|---|---|---|---|---|
| | 2 | 4 | 8 | 16 | 32 |
| stateful fusion | 67.9 | 68.4 | 66.5 | 64 | 59.3 |
| stateful NMS + stateful fusion | 68.1 | 68.6 | 66.8 | 65 | 60.6 |
| global fusion | 68.3 | 69.2 | 68 | 67.4 | 65.2 |
| stateful NMS + global fusion | **68.8** | **69.6** | **68.8** | **69.2** | **68.3** |

## 2   Single-Group vs. Multi-Group Detection Heads

We adopt single-group detection heads instead of multi-group heads. The motivation for multi-group heads is that different groups have different statistics. Based on this observation, we employ instance normalization in the center heatmap prediction and per-class NMS for single-group heads. In Tab. 2 we show that we can improve single-head detection with instance normalization and per-class NMS. The ablation studies are conducted with Cartesian pillars. We also find that when simultaneously trained with semantic segmentation, multi-group detection heads hurts segmentation accuracy. We posit the reason is, in multi-group heads some target objects are treated as background in some groups, which may confuse semantic segmentation. Considering runtime, detection and segmentation accuracy, we choose single-group head for detection.

35th Conference on Neural Information Processing Systems (NeurIPS 2021).

Table 2: Single-Group vs Multi-Group Detection Heads on the validation split of nuScenes.

| Method | instance norm | per-class nms | +seg | Runtime (ms) | Det mAP | Seg mIoU |
|---|---|---|---|---|---|---|
| Single Group | | | | 38 | 49.2 | - |
| Single Group | ✓ | | | 38 | 49.8 | - |
| Single Group | ✓ | ✓ | | 39 | 50.6 | - |
| Single Group | ✓ | ✓ | ✓ | 45 | 52.3 | 72.1 |
| Multi Group | | | | 66 | 52.1 | - |
| Multi Group | | ✓ | | 67 | 51.9 | - |
| Multi Group | | | ✓ | 73 | 52.9 | 69.8 |

Table 3: The ablation study of different #stratum following Tab. 4 in the main paper.

| det mAP / #stratum | 1 | 2 | 4 | 8 | 16 |
|---|---|---|---|---|---|
| Cartesian | 48.2 | 48.1 | 48.8 | 49.1 | 49.2 |

# 3 Ablation study on #stratum

Tab. 3 shows the effect of choosing different number of stratums for range stratified convolution and normalization. We can see a trend that adding stratum helps detection. We choose 8 stratums for the balance of performance and #paramters in our model.

# 4 Implementation

## 4.1 Network Details

The model architecture is shown in Fig.1.

## 4.2 Training Details

We use adamW [3] optimizer together with one-cycle policy [4] with LR max 0.01, division factor 10, momentum ranges from 0.95 to 0.85, fixed weight decay 0.01 to achieve super convergence. With batch size 64, the model is trained for 20 epochs.

## 4.3 Inference Details

During inference, top 1000 proposals are kept, and NMS with score threshold 0.1 is applied. Max number of boxes allowed after NMS is 83. We use NMS score threshold 0.3 for panoptic fusion.

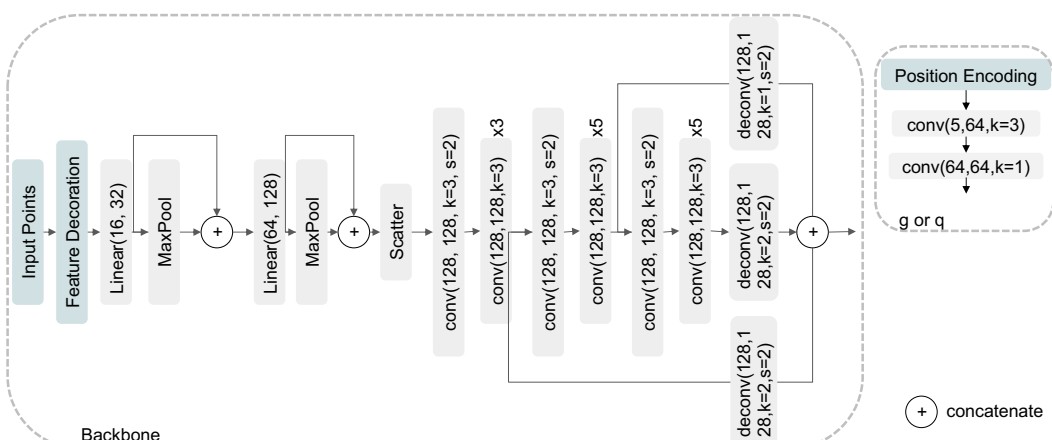

Figure 1: Network details. Feature decoration includes adding $xyz$ centers, $xyz$ clusters, $r\theta$ centers and $r\theta$ clusters.

# 5 Results on NuScenes Benchmark

## 5.1 Results on Detection Benchmark

We show detailed metrics of our methods on NuScenes 3D detection benchmark in Tab.4.

Table 4: 3D detection mAP on the NuScenes test set. We show per-class AP. mATE: average translation error; S: scale;O: orrientation; V: velocity; A: attribute. PolarStream-1:full-sweep PolarStream; PolarStream-4: PolarStream with four sectors. CPx1: trailing-edge padding; CPx2: bidirectional padding.

| Method | car | truck | bus | trailer | constr-uction vehicle | pede-strian | motor-cycle | bike | traffic cone | barr-ier | mAP↑ | mATE↓ | mASE↓ | mAOE↓ | mAVE↓ | mAAE↓ | NDS↑ |
|---|---|---|---|---|---|---|---|---|---|---|---|---|---|---|---|---|---|
| PolarStream-1 | 80.7 | 37.8 | 45.3 | 40.5 | 19.7 | 78.1 | 59.2 | 29.6 | 73.7 | 64.2 | 52.9 | 0.33 | 0.26 | 0.49 | 0.33 | 0.12 | 61.2 |
| PolarStream-4 CPx1 | 81.2 | 40.3 | 44.9 | 42 | 20.4 | 80.2 | 62.2 | 25.2 | 74.9 | 63.9 | 53.5 | 0.34 | 0.26 | 0.44 | 0.33 | 0.13 | 61.8 |
| PolarStream-4 CPx2 | 80.7 | 37.8 | 45.3 | 40.5 | 19.7 | 78.1 | 59.2 | 29.6 | 73.7 | 64.2 | 52.9 | 0.33 | 0.26 | 0.49 | 0.33 | 0.12 | 61.2 |

## 5.2 Results on Semantic Segmentation Benchmark

We show detailed metrics of our methods on NuScenes lidar semantic segmentation benchmark in Tab.5.

Table 5: Semantic Segmentation on the NuScenes test set. PolarStream-1: full-sweep PolarStream; PolarStream-4: PolarStream with four sectors. CPx1: trailing-edge padding; CPx2: bidirectional padding.

| Method | barr-ier | bike | bus | car | constr-uction vehicle | motor-cycle | pede-strian | traffic cone | trailer | truck | drive-able surface | other flat | side-walk | terr-ain | man-made | vege-tation | mIoU | freq weighted IoU |
|---|---|---|---|---|---|---|---|---|---|---|---|---|---|---|---|---|---|---|
| PolarStream-1 | 71.4 | 27.8 | 78.1 | 82 | 61.3 | 77.8 | 75.1 | 72.4 | 79.6 | 63.7 | 96 | 66.5 | 76.9 | 73 | 88.5 | 84.8 | 73.4 | 87.4 |
| PolarStream-4 CPx1 | 71.5 | 27.1 | 78.7 | 81.7 | 56.2 | 76.7 | 75.8 | 72.1 | 78.4 | 62.8 | 96.1 | 65.9 | 77 | 73.2 | 88.9 | 85.1 | 73 | 87.5 |
| PolarStream-4 CPx2 | 71.8 | 26.9 | 79.8 | 81.6 | 53.2 | 78.4 | 76.2 | 73.1 | 80.3 | 62.1 | 96.2 | 66.1 | 77 | 73.1 | 88.6 | 84.7 | 73.1 | 87.5 |

## 5.3 Results on Panoptic Segmentation

We show detailed metrics of our methods on NuScenes 3D detection benchmark in Tab.6.

# 6 Detailed Metrics

We present detailed comparison with Han et al.[1] when we slice eight sectors in a scene, shown in Tab.7. Although our PolarStream with bidirectional padding is worse than Han et al. in detection mAP, our NDS is higher because our orientation and velocity error is lower than theirs.

We also present per-class metrics of our PolarStream to see which class benefit from context the most, as shown in Tab.8 and 9. Smaller objects (in BEV) like peds and cones only see a limited improvement 1-2 points in AP while larger objects like cars, buses, trailers, construction vehicles show a bigger improvement.

# 7 Memory Usage Comparison Between Cartesian Pillars and Polar Pillars

Cuboid-shaped pillars waste computation and memory because they use larger feature maps than polar pillars. Feature map size comparsison is shown in Tab. 10. For more than 8 sectors, cartesian

Table 6: Panoptic Segmentation on the NuScenes val split. PolarStream-1: full-sweep PolarStream; PolarStream-4: PolarStream with four sectors. CPx1: trailing-edge padding; CPx2: bidirectional padding.

| Method | PQ | SQ | RQ |
|---|---|---|---|
| PolarStream-1 | 68.7 | 85.3 | 79.9 |
| PolarStream-4 CPx1 | 69 | 85.2 | 80.4 |
| PolarStream-4 CPx2 | 69.6 | 85.5 | 80.8 |

Table 7: Comparison with Han et al.(reimplemented) wrt. TP errors when there are eight sectors in a scene. CP x2: bidirectional padding. mATE: average translation error; S: scale; O: orrientation; V: velocity; A: attribute.

| Method | NDS↑ | mAP↑ | mATE↓ | mASE↓ | mAOE↓ | mAVE↓ | mAAE↓ |
|---|---|---|---|---|---|---|---|
| Han et al.[1] | 60.8 | 53.8 | 0.33 | 0.27 | 0.48 | 0.34 | 0.18 |
| Ours w/ CP x2 | 60.9 | 52.7 | 0.34 | 0.27 | 0.43 | 0.32 | 0.19 |

Table 8: Effect of Bidirectional Context Padding on nuScenes Val split for 3D Detection. CP x2: bidirectional padding.

| Method | car AP | | | | | | truck AP | | | | | |
|---|---|---|---|---|---|---|---|---|---|---|---|---|
| | 1 | 2 | 4 | 8 | 16 | 32 | 1 | 2 | 4 | 8 | 16 | 32 |
| Ours-w/o CP | 81.1 | 79.7 | 81.2 | 80.8 | 79.6 | 77.8 | 44.7 | 44.1 | 44.7 | 43.4 | 41 | 40.6 |
| Ours w/ CP x2 | - | 81.2 | 81.2 | 81.4 | 81 | 80.3 | - | 42.8 | 46 | 45.8 | 44.3 | 42.4 |

| Method | bus AP | | | | | | trailer AP | | | | | |
|---|---|---|---|---|---|---|---|---|---|---|---|---|
| | 1 | 2 | 4 | 8 | 16 | 32 | 1 | 2 | 4 | 8 | 16 | 32 |
| Ours-w/o CP | 53.4 | 55.1 | 54.8 | 54 | 52.9 | 44.6 | 29.2 | 28.7 | 28.9 | 27.2 | 25.5 | 20.2 |
| Ours w/ CP x2 | - | 54.5 | 55.6 | 55.7 | 54.4 | 51.7 | - | 28.6 | 27.6 | 29.1 | 30.2 | 26.7 |

| Method | construction vehicle AP | | | | | | pedestrian AP | | | | | |
|---|---|---|---|---|---|---|---|---|---|---|---|---|
| | 1 | 2 | 4 | 8 | 16 | 32 | 1 | 2 | 4 | 8 | 16 | 32 |
| Ours-w/o CP | 16 | 15.5 | 17.3 | 17 | 15.5 | 14.5 | 79.5 | 80.4 | 81.4 | 81.7 | 81.2 | 80.5 |
| Ours w/ CP x2 | - | 16 | 17.6 | 19.1 | 18.7 | 18.5 | - | 80.6 | 81.5 | 81.8 | 82.1 | 81.7 |

| Method | motorcycle AP | | | | | | bicycle AP | | | | | |
|---|---|---|---|---|---|---|---|---|---|---|---|---|
| | 1 | 2 | 4 | 8 | 16 | 32 | 1 | 2 | 4 | 8 | 16 | 32 |
| Ours-w/o CP | 55.5 | 55.7 | 56.9 | 57.2 | 57.7 | 53.1 | 27.9 | 28.8 | 28 | 31.3 | 35.5 | 30.4 |
| Ours w/ CP x2 | - | 57.9 | 58.2 | 60.6 | 61.6 | 58.1 | - | 28.2 | 35.2 | 36.2 | 36.2 | 36.5 |

| Method | traffic cone AP | | | | | | barrier AP | | | | | |
|---|---|---|---|---|---|---|---|---|---|---|---|---|
| | 1 | 2 | 4 | 8 | 16 | 32 | 1 | 2 | 4 | 8 | 16 | 32 |
| Ours-w/o CP | 63.3 | 65.1 | 66.7 | 67.5 | 67.2 | 65.3 | 61.2 | 62.8 | 59.1 | 62.2 | 62 | 57.4 |
| Ours w/ CP x2 | - | 65.3 | 67.7 | 67 | 68.2 | 67.1 | - | 60.2 | 60.9 | 60 | 61 | 59.6 |

pillars use twice the feature maps size as ours because half of the cartesian pillars represent empty region. Tab. 11 shows the memory usage of the feature map 'canvas' as referred to in PointPillars[2].

Table 9: Effect of Bidirectional Context Padding on nuScenes Val split for Semantic Segmentation. CP x2: bidirectional padding.

| Method | barrier IoU | | | | | | bicycle IoU | | | | | |
|---|---|---|---|---|---|---|---|---|---|---|---|---|
| | 1 | 2 | 4 | 8 | 16 | 32 | 1 | 2 | 4 | 8 | 16 | 32 |
| Ours-w/o CP | 71.5 | 70.9 | 71.1 | 70.9 | 70.9 | 69.7 | 40.3 | 36.5 | 38.2 | 37.3 | 34.2 | 29.7 |
| Ours w/ CP x2 | - | 70.9 | 70.9 | 71.5 | 71.2 | 71.2 | - | 40.8 | 42.3 | 41.6 | 43.5 | 42.9 |

| Method | bus IoU | | | | | | car IoU | | | | | |
|---|---|---|---|---|---|---|---|---|---|---|---|---|
| | 1 | 2 | 4 | 8 | 16 | 32 | 1 | 2 | 4 | 8 | 16 | 32 |
| Ours-w/o CP | 86.4 | 88.9 | 83.2 | 81.2 | 79 | 74.9 | 87.7 | 85.2 | 86.6 | 81.9 | 82 | 80.7 |
| Ours w/ CP x2 | - | 86.8 | 87.8 | 87.6 | 86.7 | 86.7 | - | 83.5 | 86.1 | 83.4 | 82.5 | 83 |

| Method | construction vehicle IoU | | | | | | motorcycle IoU | | | | | |
|---|---|---|---|---|---|---|---|---|---|---|---|---|
| | 1 | 2 | 4 | 8 | 16 | 32 | 1 | 2 | 4 | 8 | 16 | 32 |
| Ours-w/o CP | 54.4 | 50.9 | 46.7 | 48.1 | 43.3 | 37.3 | 80 | 78.3 | 78.8 | 67.4 | 72.1 | 69.2 |
| Ours w/ CP x2 | - | 51.1 | 52.5 | 52.2 | 51.5 | 47.7 | - | 77.3 | 80.2 | 79.8 | 80.8 | 77.9 |

| Method | pedestrian IoU | | | | | | traffic cone IoU | | | | | |
|---|---|---|---|---|---|---|---|---|---|---|---|---|
| | 1 | 2 | 4 | 8 | 16 | 32 | 1 | 2 | 4 | 8 | 16 | 32 |
| Ours-w/o CP | 79.4 | 78.9 | 78.2 | 79.2 | 76.5 | 74.3 | 65.6 | 66.3 | 65.6 | 64.2 | 63.5 | 60.8 |
| Ours w/ CP x2 | - | 78 | 79.1 | 80 | 80.7 | 80.1 | - | 66 | 67.1 | 66.3 | 66.5 | 67.1 |

| Method | trailer IoU | | | | | | truck IoU | | | | | |
|---|---|---|---|---|---|---|---|---|---|---|---|---|
| | 1 | 2 | 4 | 8 | 16 | 32 | 1 | 2 | 4 | 8 | 16 | 32 |
| Ours-w/o CP | 66.8 | 64.4 | 59.2 | 54.2 | 48.7 | 45.5 | 78.5 | 77.9 | 75 | 68.3 | 68.3 | 61.5 |
| Ours w/ CP x2 | - | 61.1 | 63 | 62.5 | 62.5 | 60.3 | - | 76.9 | 75.9 | 76.4 | 76.3 | 75.3 |

| Method | driveable region IoU | | | | | | other flat IoU | | | | | |
|---|---|---|---|---|---|---|---|---|---|---|---|---|
| | 1 | 2 | 4 | 8 | 16 | 32 | 1 | 2 | 4 | 8 | 16 | 32 |
| Ours-w/o CP | 95.8 | 95.6 | 96.6 | 95.6 | 94.9 | 94.9 | 65.9 | 66.8 | 68.4 | 67.7 | 64.5 | 63.4 |
| Ours w/ CP x2 | - | 95.3 | 95.7 | 95.7 | 95.6 | 95.5 | - | 67.8 | 73 | 66.4 | 66.8 | 67.6 |

| Method | sidewalk IoU | | | | | | terrain IoU | | | | | |
|---|---|---|---|---|---|---|---|---|---|---|---|---|
| | 1 | 2 | 4 | 8 | 16 | 32 | 1 | 2 | 4 | 8 | 16 | 32 |
| Ours-w/o CP | 73.2 | 73.1 | 72.9 | 72.1 | 70.8 | 69.3 | 72.6 | 72.6 | 72.7 | 72.7 | 72 | 71.5 |
| Ours w/ CP x2 | - | 73.1 | 73 | 72.8 | 72.6 | 71.6 | - | 72.7 | 72.8 | 72.6 | 71.5 | 71.1 |

| Method | manmade IoU | | | | | | vegetation IoU | | | | | |
|---|---|---|---|---|---|---|---|---|---|---|---|---|
| | 1 | 2 | 4 | 8 | 16 | 32 | 1 | 2 | 4 | 8 | 16 | 32 |
| Ours-w/o CP | 88.4 | 88.1 | 88 | 87.7 | 87.5 | 87 | 84.2 | 83.5 | 83.8 | 83.7 | 83.4 | 83.7 |
| Ours w/ CP x2 | - | 88.4 | 88.2 | 86.7 | 88 | 83.9 | - | 84.4 | 84.3 | 84.2 | 84.1 | 83.9 |

Table 10: Input feature map size comparison using different coordinate system.

| Coordinate System / #sectors | 1 | 2 | 4 | 8 | 16 | 32 |
|---|---|---|---|---|---|---|
| Cartesian | 512x512 | 512x256 | 512x128 | 512x128 | 512x64 | 512x32 |
| Polar | 512x512 | 512x256 | 512x128 | 512x64 | 512x32 | 512x26 |

Table 11: Memory usage comparison using different coordinate system. We report the memory usage for 'canvas' as referred to in PointPillars[2]. The memory is per sector in MB.

| Coordinate System / #sectors | 1 | 2 | 4 | 8 | 16 | 32 |
|---|---|---|---|---|---|---|
| Cartesian | 33.6 | 16.8 | 8.4 | 8.4 | 4.2 | 2.1 |
| Polar | 33.6 | 16.8 | 8.4 | 4.2 | 2.1 | 1.3 |