# OpenReview forum: "PolarStream: Streaming Object Detection and Segmentation with Polar Pillars"
_NeurIPS.cc/2021/Conference — NeurIPS 2021 Poster_

### Official Review · Reviewer_2ez1 · 2021-07-15

**Rating:** 6
**Confidence:** 4

**Summary:**

This paper studies an interesting and important problem: streaming-based 3D perception. The frequency of LiDAR scans are usually much lower than RGB cameras (10 FPS vs 30 FPS), and the speed of non-streaming LiDAR detectors will be limited to 10 FPS. The proposed approach can achieve higher FPS than the LiDAR frequency thanks to the sector-by-sector processing with PolarPillars. It utilizes multi-scale context padding to deal with the border effects, and addresses the distortion artifacts in object detection via Feature Undistortion + Range Stratified Convolution&Normalization. The proposed method achieves state-of-the-art performance (among 2D projection-based methods) on streaming-based 3D semantic segmentation and object detection on the nuScenes dataset, and the results are even comparable with non-streaming methods.

**Limitations And Societal Impact:**

The main limitation of this work is that the proposed method is evaluated only on the nuScenes dataset (though it is hard to find other datasets that have all 3 kind of tasks in this paper).

**Main Review:**

This paper attempts to improve the efficiency of real-world 3D perception through a streaming-based pipeline. The authors propose novel techniques such as feature undistortion and range stratified convolution & normalization to address the distortion caused by polar projection, which is simple and effective. The padding strategy takes advantage of temporal information and refines the border region, improve the performance of PolarPillars under large # of sectors significantly. This work also presents a solution to jointly do semseg, panopticseg and object detection with a single network, which can also improve the efficiency of 3D perception.

For the experiment evaluation, I believe the results for PolarPillar is much better than simple projection-based methods (such as PolarSeg and PointPillars). Achieving almost no accuracy loss at #sectors = 32 is impressive. It will be better if the authors can also provide batch size = 1 latency of PolarPillars at different #sectors in a table. I also wonder whether there's some chance that we can integrate SparseConv-based feature extractors into PolarPillars in the streaming mode. Potential experiment results on detection only will be very helpful. In the future the authors can also try extending PolarPillars to other datasets such as Waymo (detection) and SemanticKITTI (semseg + panopticseg).

There's a minor concern on the end-to-end latency calculation for PolarPillars. I'm not sure whether the authors follow the standard protocol in nuScenes detection to use 10-sweep point cloud as the input. If so, did you consider the time loading these point clouds, and what is the proportion of the data loading time?

Overall, I believe this work has solid contributions and good quality.




**Time Spent Reviewing:**

1.5

---

> ### Author Response · Authors · 2021-08-10
> **latency table added; results of detection only added; questions clarified**
>
> We sincerely thank the reviewer for the constructive comments and hope our responses can address the concerns
>
> -  batch size = 1 latency of PolarPillars at different #sectors in a table.
>
>
>
>
> | #sectors | lidar spin latency /ms | polarstream latency /ms |end-to-end latency /ms|
> | -------- | -------- | -------- | -------- |
> | 1   |50    | 44.9    | 94.9 |
> | 2| 25|27.3|52.3|
> | 4| 12.5 | 22.8 | 35.3 |
> | 8 | 6.3 | 19.2 | 25.5 |
> | 16 | 3.1 | 16.1 | 19.3 |
> | 32 | 1.6 | 12.6 | 14.2 |
>
> also refer to Figure 1. Note that polarstream latency does not scale down linearly as #sectors increases. The amount of computation scales linearly as #sectors increases but latency is more complicated. It also depends how parallel the computation is, which depends on if the gpu memory is saturated. This is also observed in Han et. al.[13] Figure 5.
>
> - whether there's some chance that we can integrate SparseConv-based feature extractors into PolarPillars in the streaming mode.
>
> The reviewer raises a good point. Our model is compatible with all voxel-based method. It’s absolutely feasible to integrate SparseConv especially for low-level features and save more memory.
>
> - Potential experiment results on detection only will be very helpful.
>
> We tried detection only for 32 sectors and our method got detection map at 50, compared to 48.7 for Han et al., which is consistent to results in Table 1.
>
> - minor concern on the end-to-end latency calculation for PolarPillars
>
> Using 10-sweep point cloud is a common practice as in CenterPoint[35], HotSpotNet[6], CVCNet[5], PointPainting[29], PointPillars[15]. 1-sweep point cloud is sparser and leads to inferior detection results especially in velocity estimation compared to 10-sweep (mAP 46.7 vs 50.6).
>
> We did not consider point clouds loading time because we assume it is immediately accessible onboard from sensor. Plus, since all methods use 10-sweep data, data loading is not a variant we care about. Rather, we take data warping time, the time of transforming points from previous nine frames to current frame, into account. In our comparison, STROBE does not need data warping, while all other methods including our methods do.  So data warping time is a variant we care about. And data warping time for full-sweep is 2ms,  approximate 4.4% of our full-sweep polarstream latency.

---

> > ### Comment · Reviewer_2ez1 · 2021-08-12
> > **Thanks for the response.**
> >
> > Thanks for the response. I'm positive towards the acceptance of this paper.

---

### Official Review · Reviewer_8rdp · 2021-07-16

**Rating:** 7
**Confidence:** 5

**Summary:**

This paper proposes a new "PolarStream" network architecture for lidar detection task. Stream-based lidar inputs are fed into polar-coordinate voxelizer. Several new techniques are proposed in order to improve detection quality: Multi-scale context padding, Feature Undistortion, Range Stratified Convolution & Normalization.


**Limitations And Societal Impact:**

I don't think there is any "limitations and potential negative societal impact" in this work. This is a very general lidar/camera detection task/model.

**Main Review:**

I think the method this paper proposes is novel, and increamental improvements are explained in detail. Therefore I am giving it an accept.


**Time Spent Reviewing:**

2.5 hours

---

> ### Author Response · Authors · 2021-08-09
> **Thank you**
>
> We sincerely thank the reviewer for recognizing our contributions and appreciation of the paper.

---

### Official Review · Reviewer_88yB · 2021-07-20

**Rating:** 6
**Confidence:** 3

**Summary:**

This work proposes a method of unwarping lidar sectors obtaining in a streaming setting (i.e., without waiting for a complete sweep to complete) in the (r,theta) (polar) space, and using standard convolution on top.  A new “contextual” padding scheme using ego-motion-compensated previous scan-data is proposed. Further, two other innovations — (i) feature undistortion to undo the effect of polar warping, and (ii) stratified processing (for differing statistics of data with distance) are proposed.

Experiments for detection, segmentation, and panoptic segmentation are carried out on the nuScene dataset. There is insufficient empirical evidence to establish the place of the proposed method w.r.t previous methods (see “quality” below).

**Ethical Concerns:**

No.

**Limitations And Societal Impact:**

No.

### Suggestions:
Applications of lidar detection and segmentation could be in sensitive areas like autonomous driving, and autonomous weapons. Hence, errors must be adequately quantified and presented (for driving), and access to performant lidar processing algorithms be monitored.

**Main Review:**

### Originality
It is difficult to place what contributions are novel aspects of this work. This is because a number of methods for using cylindrical, spherical and hybrid coordinate systems have been proposed already [1,15,36,37,38,40] – the relation and distinction w.r.t to these methods has not been discussed.

The use of unwarped sectors (in (r, theta) space) in a streaming setting + the contextual padding seem to be the key original contributions — which might have limited novelty.
&nbsp;

### Quality
The empirical evaluation seems to be lacking:

1. The baseline methods in Table 1 have not been validated on the original datasets — so it is difficult to place the results in context. Were the baselines implemented correctly? The proposed method has only been evaluated on nuScenes — again limiting fair comparison with previous methods (e.g., with Han et al. who evaluate on Waymo open dataset).

2. In the comparison on the nuScenes dataset in Fig. 5: while the method is faster it lags behind existing methods in performance. It could be that the proposed method has traded-off accuracy with latency, e.g., by using a smaller backbone.  The exact details of the backbones and number of parameters in the baseline methods report are not made available – hence, it is again difficult to place these results.
&nbsp;

### Clarity
1. This work builds on an existing body of methods for detection and segmentation in lidar data. The text defers the discussion of these details to references. This makes it difficult to grasp the proposed method adequately. Hence, the writing can be improved by simplifying the description, making the text more self-contained.

2. The experiments section can be improved by collecting the “effect of multi-scale padding” (L259) and “ablation” (section 5.5) together.
&nbsp;

### Significance
Per Table 1, for small ‘n’ i.e., when large sectors are used, the detection quality of the proposed method is comparable, while the segmentation accuracy is marginally higher than the baselines.  Only for small sectors does the proposed contextual padding from previous scan data aids the performance. Hence, one of the key contributions of this work seems to be the contextual padding (as also evidenced by long discussion devoted to padding – L259-267, L294-309, fig. 4), which is a novel but incremental contribution.

Further, the authors themselves have implemented the baseline methods (in Table 1), without validating on the original datasets that the baselines were evaluated on (e.g., Waymo open dataset). Hence, it is difficult to place the reported results in context. It is unclear if the baseline methods were trained with appropriate hyper-parameters etc.

Consequently, the significance of this work might be limited.
&nbsp;

### Clarifications / Typos:
1. L159: “we model $w’_k$ as …”: “as” is missing?
2. Fig. 2:  Unclear where “unfold in BEV” is given as input? To “Pillar Feature Encoder”? Please clarify.
3. L129: “mulit-scale” → multi-scale.
4. How is the *independent* motion of objects (not ego-motion of the lidar) in the previous/old sector compensated (or not) for context padding? Seems like it is not, as there is no description to this end. Would this not introduce inaccuracies in the point-cloud? Please clarify.
5. How is the ego-motion determined for bi-directional context padding (L202)?
6. Is the piller-size on L207 dependent on ‘n’? Or is it held fixed, as ‘n’ changes in the experiments?
7. In Table 1, why is there a significant drop in performance in the detection metrics when going from 16 to 32 sectors?


**Time Spent Reviewing:**

4

---

> ### Author Response · Authors · 2021-08-10
> **our model with a heavier backbone outperformed previous SOTA in semantic segmentation; explain reimplemented baselines; clarified novelty and other questions**
>
> We thank the reviewer for the review and insightful comments. We have addressed the points raised by the reviewer below:
>
> - contribution and novelty
>
> Thanks for raising this point. This paper focuses on improving object detection from a stream of lidar data  -- a key component of a low latency real time detection system for AVs. Our starting point is the insight that a polar coordinate system is ideally suited to this problem domain. Based on this we develop the first polar coordinate based streaming architecture. Contributions include addressing the context issue inherent in streaming based architectures; establishing baselines (with open source code) on a public benchmark; and a thorough exploration of how other techniques such as range-stratified convolutions help benefit the design.
>
> - The baseline methods in Table 1 have not been validated on the original datasets
>
> The reviewer raises a great question here. We were unable to reproduce previous work as detailed below and we consider it a key contribution of our work to share an open-source implementation of the previous methods as well as full implementation protocol for a streaming version of the nuScenes dataset.
>
> So why were we not able to reproduce previous work? First reason is straight-forward: previous work did not release code so we were unable to run them on the nuScenes dataset. Second: previous work did not run on open source data. This is especially clear for STROBE which published on a private dataset. But it is also true for Han et al. While they evaluated on the Waymo open dataset, it was in fact evaluated on a *simulated streaming version* thereof. Critically, there are many ways to simulate streaming and we were unable to get a hold of the authors to obtain the required details. For that reason re-running our implementation of Han et al. on our version of a streaming version of Waymo Open dataset would not have sufficed to establish implementation correctness.
>
>
> - In the comparison on the nuScenes dataset in Fig. 5: while the method is faster it lags behind existing methods in performance. It could be that the proposed method has traded-off accuracy with latency, e.g., by using a smaller backbone. The exact details of the backbones and number of parameters in the baseline methods report are not made available – hence, it is again difficult to place these results.
>
> The reviewer correctly points out that there is a latency vs accuracy tradeoff and we concede that Figure 5 is a bit misleading. To recap: our main claim is that our method is better than other streaming methods, as shown in Table 1. Since there are so few streaming based baselines, and as mentioned in previous reply, they are neither open source nor reproducible, we also compare our results to non-streaming methods in Figure 5.  The non-streaming methods use different size backbones and encoders. Below we show that we can recover state of the art detection accuracy if we use a larger backbone in our method.
>
> We also tried our full-sweep polarstream (PLS1) with the same backbone as in CenterPoint (named PLS1 heavy in the following table) and the following table is what we got for detection and semantic segmentation on NuScene validation set. CenterPoint and Cylinder3D are the SOTA methods for detection and semantic segmentation respectively. Our PLS1 heavy is able to match/beat the SOTA models for detection (CenterPoint) and segmentation (Cylinder3D) on the nuScenes validation set.  In our work we focus on onboard applications like streaming and those heavy backbones cannot run onboard.  So we choose the encoder and backbone from the PointPillars [15] model. We provided the details of the backbone in supplementary Figure 1.
>
> | Methods | det mAP | seg mIoU | runtime | #parameters|
> | -------- | -------- | -------- | -------- | -------- |
> | CenterPoint    | 56.4     |      | 11Hz   | 149MB|
> | Cylinder3D | | 76.1 |11Hz(reported) 2Hz(reproduced)| 215MB|
> | Ours-PLS1 | 51.2| |26Hz|65MB|
> |Ours-PLS1 | |73.8|34Hz|65MB|
> |Ours-PLS1 heavy|56.2| |11Hz|149MB|
> |Ours-PLS1 heavy||76.8| 15Hz|149MB|
>
> - Fig. 2: Unclear where “unfold in BEV” is given as input? To “Pillar Feature Encoder”? Please clarify.
>
> Thanks for the feedback; we will make this more clear in revision. The input is a Wedge. In more details: In the BEV, the polar pillars form a wedge-shaped region on x-y plane, but convolution requires rectangular grid-structure. We therefore need to unfold the wedge-shape input region on x-y plane to a rectangular input region on r-theta plane. One dimension is r (range) and the other dimension is theta (azimuth).
>
> - How is the ego-motion determined for bi-directional context padding
>
> For the corresponding input point P_t=[x_t, y_t, z_t, 1] at time t and P_t’=[x_t’, y_t’, z_t’, 1] at time t’, the ego-motion matrix from t to t’ is M, where M is a 4x4 matrix and P_t’ = MP_t. M is a known matrix from driving logs.  For context padding, we pad the features on BEV so it is only 2D. Say the corresponding features are F_t at [x_t, y_t, 1] F_t’ at [x_t’, y_t’, 1] from time t to t’. We get 2D ego-motion M_2d by extracting the first 3x3 block from M. So M_2d is also a known 3x3 matrix. Then we use M_2d to warp feature map from time t to t’.
>
> - Is the pillar-size on L207 dependent on ‘n’? Or is it held fixed, as ‘n’ changes in the experiments?
>
> it is fixed.
>
> - In Table 1, why is there a significant drop in performance in the detection metrics when going from 16 to 32 sectors?
>
>
> As the reviewer points out  the performance drops across the board when going from 16 to 32 sections. We believe it has to do with receptive field.  16 sectors means pi/8 rads for a sector, which at 10m range corresponds to a span of pi/8*10=3.9m -- large enough to cover a vehicle. But for 32 sectors, at 10m range, it corresponds to a span of 1m, not enough to cover a vehicle.
> For our methods, det map for 32 sectors is 52.8 instead of 51 (training job was not finished by the time of submission). So our gap from 16 sectors to 32 sectors is smaller than other methods (ours -1.4, Han et. al. -2.1 and STROBE -3.7), which again shows the advantage of context padding.

---

> > ### Comment · Reviewer_88yB · 2021-08-31
> > **After Authors' Response**
> >
> > Thank the authors for their response to the concerns raised in the review.
> >
> > Given the clarification regarding non-availability of open baseline implementations, and the attempt of the authors to standardize and release these ("good citizenship", as highlighted by reviewer ijm6),  I am inclined to update my rating positively from "below acceptance" threshold to "above".

---

### Official Review · Reviewer_ijm6 · 2021-07-22

**Rating:** 6
**Confidence:** 3

**Summary:**

This work improves the accuracy and efficiency of streaming lidar processing for visual recognition by adopting polar coordinates. Streaming lidar sectors are slices of a circle, so switching to polar coordinates is more efficient because angles and ranges align with these slices. Existing methods based on cartesian coordinates must instead process the containing rectangles of the slices. Having switched to polar coordinates, this work introduces a context representation that pads sectors with following/preceding sectors, making use of their now compatible spatial coordinates in the polar representation. Polar processing is not without its drawbacks though for standard convolutional processing, so tweaks are proposed to (1) fix feature distortion and (2) deal with size changes w.r.t. range. The combination of polar processing, contextual padding, and these polar tricks or fixes is the contributed PolarStream method. Conceptual figures illustrate the polar/cartesian difference and its implications for incorporating context and its impact on planar convolutional filtering. Experiments report (1) comparison with other streaming methods showing better accuracy/latency, (2) comparison with the state-of-the-art or near it at the time of submission for all methods showing better latency but worse accuracy, and (3) ablations of the proposed tweaks on planar convolutional filtering from polar inputs that show they indeed help polar processing and are not confounded with other effects on cartesian processing. The role of context and its implementation in prior work and this work is discussed in the text and illustrated by figures in a unifying way that could inform follow-up improvements of streaming lidar processing.

**Ethical Concerns:**

None.

**Limitations And Societal Impact:**

The limitations are not addressed in their own section, but the discussion and comparison with existing methods identifies directions for improvement. The ablations (Sec 5.5) against cartesian vs. polar pillars are adequate for this purpose.

No particular societal impacts are raised by this paper more than by any other on autonomous driving.

**Main Review:**

Novelty
- Streaming lidar processing is already established by [11, 13] as acknowledged.
- This work switches to a polar coordinate system while existing work uses cartesian coordinates.
- The polar coordinate approach then allows for a new kind of multi-scale context (called bidirectional padding by this work) and tricks to improve convolutional processing of the polar inputs (feature undistortion and range stratification).
  - Feature undistortion maps from polar coordinates back to cartesian coordinates for convolutional filtering. It is like an inverse of other input/coordinate transformation methods, like warped convolution, but differs in that its exact transformation is learned as a fully convolutional network.
  - Range stratification divides convolution and normalization across ranges with distinct parameters and statistics. This is not unlike domain-specific normalization (AdaBN), but in this case it is uniquely spatially-motivated by the use of polar coordinates.
- All-in-all the proposed changes are a new refinement of streaming/sector-wise lidar processing.

Significance
- The proposed polarstream is 3-5x faster than the most accurate methods, which is a significant reduction in latency for time-critical tasks like perception for autonomous driving. However, the most accurate methods are 5-15 points absolute better in detection (mAP) and segmentation (IoU) performance. Among methods of comparable latency, the proposed method is the more accurate by a similar margin, making it state-of-the-art among streaming methods.
- Only methods reporting accuracy and runtime are compared against. While latency is of particular importance for this application, it would be informative to report the most accurate method (regardless of runtime) for completeness.
- The detection and segmentation metrics deserve more weight, as the panoptic metric is not based on dataset-provided instance labels (Section 5.1). The proposed method does improve on the detection and segmentation metrics.

Clarity
- The main baselines for streaming are [11,13] but both are reproduced due to lack of code. Including the performance of the reproduction vs. the original results on the datasets reported in the papers would build confidence in the reproductions.
- Detailed discussion of prior work and failure modes (Sec. 5.3, Figure 4, ...) help identify the difference in methods and isolate the various effects of sectors and how to process them. I found this more educational than simply reporting absolute numbers to know what is better and worse.
- Ablations check the suitability of the proposed components, feature undistortion and range stratified conv + norm, for streaming/sector methods. These both help the polar representation, but not cartesian representation, showing the effect is due to the needs of sector-based streaming processing and not some confound like parameter count.
- The implementation section needs an optimization subsection, or a reference to the supplement, to cover training hyperparameters.

Related Work
- The provided references and discussion are sufficiently complete and detailed.
- To give more context on (log)-polar processing in deep learning for vision, consider mentioning Warped Convolutions by Henriques & Vedaldi ICML'17 and polar transformer nets by Esteves et al. ICLR'18.
- Although different, existing work on hybridizing convolution and range like SurfConv by Chu et al. CVPR'18 or Range Sparse Net by Sun et al. CVPR'21 (after submission deadline) could be relevant enough for discussion.

Decision
This work is borderline due to the large accuracy gap between non-streaming and streaming methods and the not entirely certain reproduction of the baselines. However, the proposed methods for polar processing of streaming sectors seem to (1) better respect the structure of the data as arcs/polar sweeps and (2) improve accuracy and efficiency over existing streaming methods.

For Rebuttal
- Please comment on the accuracy of polarstream vs. the state-of-the-art for accuracy metrics. Is it striking a good balance between accuracy and latency? Figure 5 is about this, but specifically mentions that only methods with runtime are reported—is there a more accurate method that is excluded?
- Please explain how polarstream with bidirectional padding improves in mAP with more sectors, at least in the range 2-16? This seems surprising, and diverges from results for the other tasks. (Sec. 5.3 gives a hypothesis, but can this not be measured to check?)
- Please gauge the fidelity/quality of the reproduced baselines in the context of their originally reported results, to ensure the new results are good indicators w.r.t. existing methods.

Other Comments
- Consider plotting in alternative colors than red/green for colorblindness. Blue/orange is a substitute in some textbooks, for instance.
- Given the focus on lidar and autonomous driving, with contributions on how to structure the data and processing for the sensor, would this work have more impact at CVPR/ECCV/ICCV or IV (intelligent vehicles)?

**Time Spent Reviewing:**

3

---

> ### Author Response · Authors · 2021-08-10
> **our model with a heavier backbone ourperformed previous SOTA in semantic segmentation; explaination on reproduction of baselines and why streaming detection outperforms full-sweep**
>
> We thank the reviewer for their insightful comments and time spent reviewing the paper.
>
> - the large accuracy gap between non-streaming and streaming methods. Please comment on the accuracy of polarstream vs. the state-of-the-art for accuracy metrics.
>
> The gap comes from the fact that we used a lighter backbone as in PointPillars which operates on the pillar feature encoder while others use the much slower ResNet-like backbone on top of 3D voxel encoder. It is not a gap between streaming and non-streaming methods.
> We also tried our full-sweep polarstream with the same 3D ResNet-like encoder as in CenterPoint (named PLS1 heavy in the following table) and we were able to match/beat the SOTA models for detection (CenterPoint) and segmentation (Cylinder3D) on the nuScenes validation set. In our work we focus on onboard applications like streaming and those heavy backbones cannot run onboard. So we choose the encoder and backbone from the PointPillars [15] model.
>
>
>
> | Methods | det mAP | seg mIoU | runtime | #parameters|
> | -------- | -------- | -------- | -------- | -------- |
> | CenterPoint    | 56.4     |      | 11Hz   | 149MB|
> | Cylinder3D | | 76.1 |11Hz(reported) 2Hz(reproduced)| 215MB|
> | Ours-PLS1 | 51.2| |26Hz|65MB|
> |Ours-PLS1 | |73.8|34Hz|65MB|
> |Ours-PLS1 heavy|56.2| |11Hz|149MB|
> |Ours-PLS1 heavy||76.8| 15Hz|149MB|
>
> - the not entirely certain reproduction of the baselines. Please gauge the fidelity/quality of the reproduced baselines in the context of their originally reported results, to ensure the new results are good indicators w.r.t. existing methods.
>
> Previous methods did not release their code and they either worked on a private dataset or the dataset details are not released. Since there is no streaming dataset available to us, we decided to make a streaming benchmark out of NuScenes and put all methods under the same condition for comparison. We make dataset, backbone, augmentation, optimization all the same in order to focus on how to address limited spatial view in streaming sectors. The benefit of using NuScenes dataset is there are ten classes of different sizes so we can observe how limited spatial view affects objects of different sizes (as shown in table 7&8 in Supplementary).
>
> We tried our best to get the best results out of reimplemented methods and we got feedback from the authors of STROBE for implementation details. We also emailed the authors of Han et. al. but did not get reply. We will release the code for streaming dataset, our method and our reimplemented methods so everyone can base their work on ours. Previous methods not releasing their code should not block further research. If it is a blocker for our paper, it will also block any following paper on streaming. Rather, we will make everything public to facilitate further research, which is also one of our contributions.
>
>
>
> - Please explain how polarstream with bidirectional padding improves in mAP with more sectors, at least in the range 2-16? This seems surprising, and diverges from results for the other tasks. (Sec. 5.3 gives a hypothesis, but can this not be measured to check?)
>
> Streaming based object detection (2-16 sectors) consistently does better than full sweep object detection while semantic segmentation does not follow this trend. The difference between anchor-free detection and semantic segmentation is that detection also requires localization of the bounding boxes. We find that the Average Orientation Error (AOE) is consistently lower for models operating on 2-16 sectors than full sweep (0.41 rad vs 0.44 rad). This adds more weight to our hypothesis in section 5.3 and bounding box regression becomes easier for streaming perception.

---

> > ### Comment · Reviewer_ijm6 · 2021-08-21
> > **Thank you for the thorough response.**
> >
> > > The gap comes from the fact that we used a lighter backbone
> > > It is not a gap between streaming and non-streaming methods
> >
> > Thank you for the clarification on the difference in backbones as the driver of the gap. This could be mentioned in the text, or at least mentioned in the supplement and referenced from the experiments section.
> >
> > > Previous methods did not release their code and they either worked on a private dataset or the dataset details are not released.
> >
> > This obstacle is obviously out of the hands of this submission, and is in fact an issue any research on streaming lidar processing. In the absence of existing public materials, providing a reproducible dataset and method for this setup is indeed a contribution.
> >
> > I am positive on acceptance of this submission, because its technical and empirical content, as well as its "good citizenship" bonus of making research more public and reproducible by the release of the dataset and code in contrast to prior work with private code and closed or undocumented data.

---

### Decision · Program_Chairs · 2021-09-28

**Decision:**

Accept (Poster)

**Comment:**

4 expert reviewers suggest acceptance after rebuttal. This is a good quality paper making valuable contributions for streaming lidar-based object localization and that is promised to provide an open benchmark and codebase for the community.

**Consistency Experiment:**

NeurIPS has a long history of experimentation. In 2014, NeurIPS ran an experiment in which 10% of submissions were reviewed by two independent committees to quantify the randomness in the review process. This year, we repeated a variant of this experiment to see how the quality of the review process has changed over time.  This paper was part of the experiment and was therefore assigned to two committees (consisting of reviewers, an Area Chair, and a Senior Area Chair) that reached independent decisions.  If both committees made the same recommendation, this recommendation was followed. If a single committee recommended acceptance, the paper was accepted (with the exception of a few cases in which the other committee identified what we considered a fatal flaw, e.g., an error in a key result).

This copy’s committee reached the following decision: **Accept (Poster)**

The other committee assigned to the paper recommended **Reject**.  You can find the other set of reviews, along with any follow up discussion with the authors here:
https://openreview.net/forum?id=lzGLd1nKb7Y